# Validation of the Opening Minds Scale for Health Care Providers (OMS-HC): Factor Structure and Psychometric Properties of the Brazilian Version

**DOI:** 10.3390/healthcare11071049

**Published:** 2023-04-06

**Authors:** Bruna Sordi Carrara, Marcos Sanches, Sireesha Jennifer Bobbili, Simone de Godoy Costa, Álvaro Francisco Lopes de Sousa, Jacqueline de Souza, Carla Aparecida Arena Ventura

**Affiliations:** 1PAHO/WHO Collaborating Centre for Nursing Research Development-Brazil, University of São Paulo at Ribeirão Preto College of Nursing, Ribeirão Preto 14040-902, Brazil; 2Centre for Addiction & Mental Health, Toronto, ON M6J 1H4, Canada; 3Instituto de Ensino e Pesquisa, Hospital Sírio-Libanês, São Paulo 01308-060, Brazil

**Keywords:** factorial structure, psychometric properties, stigma, health professionals

## Abstract

Stigma towards people with mental illness is also present among health professionals. The study validated and estimated the reliability, dimensionality and structure of the Opening Minds Stigma Scale for Health Care Providers (OMS-HC) scale in Brazil. In this methodological study, health professionals (n = 199) from Family Health Units in Brazil were recruited by convenience sampling. The EFA conducted with 16 items resulted in four factors. The Cronbach’s Alpha for the OMS scale was 0.74, which is considered to reflect reasonable reliability. The data presented contribute to the use of the scale in studies that investigate the level of stigma among health professionals towards people with mental illness, as well as in the development of anti-stigma interventions in this context.

## 1. Introduction

Of all Latin American countries, Brazil has the highest prevalence of mental illness among adults aged 15 to 59 years, with elevated rates of anxiety disorders, mood disorders and substance use disorders [1]. In general, people with mental illness are perceived as strange, frightening, unpredictable, aggressive and lacking self-control [2,3,4]. Stigma towards people with mental illness is associated with these negative perceptions, which have an impact on people of all ages, cultures, and socioeconomic conditions [5].

Stigma may act as a barrier to accessing health services [6] and can significantly affect the potential for recovery, reinforcing negative attitudes and behaviors towards people with mental illness [7]. Health care providers, like the general population, are no less susceptible to stigmatizing beliefs and behaviors than the general public, and evidence demonstrates that stigma and discrimination are prevalent in health care settings [8,9]. Research also shows that stigma conveyed by health care providers is associated with poor help-seeking behaviors and low adherence to mental health treatments among those living with a mental illness [10,11,12].

Notably, 3% of the Brazilian population requires continuous mental health care due to serious mental illness, while 9% of the population requires constant care due to common mental illness, placing a high demand on Primary Health Care Services (PHC services) [13]. Since stigma is present among health care professionals [14,15] in Brazil, these individuals are in a unique position to act as powerful agents of de-stigmatization by providing humanizing, welcoming treatment that facilitates social reintegration and a recovery-based approach to treatment [16,17]. In Brazil, Primary Health Care (PHC) is administered through the Family Health Strategy (FHS) and is the preferred care pathway to access Single Health System (SUS) services, the Brazilian public health system created by the Federal Constitution of 1988. Thus, PHC is a great venue for inciting change related to stigma among health care professionals [18,19]. 

Considering the detrimental effects of stigmatization, it is important to investigate this phenomenon in various sociocultural contexts to develop and test effective interventions [20,21]. It is also essential to involve health care providers in the research process to ensure their perspectives are included and interventions actively challenge stigmatizing beliefs and attitudes, which facilitate productive relationships with people with mental illness [22].

The Opening Minds Scale for Health care Providers (OMS-HC) was developed in Canada within the scope of the “Opening Minds” anti-stigma initiative to determine the degree of stigma exerted by health professionals towards people with mental disorders, with the aim of assessing the effectiveness of anti-stigma programs in reducing the impact of stigma on care delivery [23,24]. In this sense, the OMS-HC was developed for use in the Canadian context and designed to use few items to assess a set of factors related to stigma, making its use practical in the evaluation of anti-stigma interventions [25].

OMS-HC is a widely used self-report questionnaire that assesses several dimensions of health care professionals’ stigmatizing attitude towards their patients with mental health problems [25]. The initial version of the scale consisted of 20 items, and its factorial structure showed a two-factor solution with 12 items that seemed to lack the important social distance dimension of the stigma construct. 

The psychometric properties of the scale have also been tested in international studies, being a reliable and valid scale in Singapore [26], Italy [27] and Chile [28] in several populations of health professionals. 

In Brazil, a research group is developing the same anti-stigma intervention as used by Primary Health Care professionals in Canada, before which only the cultural adaptation of the OMS-HC had been carried out, which is not sufficient, because the psychometric properties of the instrument were not evaluated. The choice of the OMS-HC version with 20 items is due to the fact that it was the version used in the evaluation of the effectiveness of the Canadian anti-stigma intervention. In addition, in Brazil, we started applying all 20 items to ensure that no item was missing that could eventually be important in the specific Brazilian population. Thus, the factorial analysis resulted in the use of only 16 items, showing that the original version of the scale does not seem to work so well in Brazil, thus evidencing the importance of the study through the evaluation of the psychometric properties of the OMS-HC used in Brazil.

Unfortunately, in Brazil, there are limited validated and culturally relevant instruments available to measure the mental illness- and substance use-related stigma phenomenon among health professionals. To address this gap and expand the possibilities of reducing stigma, the present study seeks to examine the relevance of the Opening Minds Stigma Scale for Health Care Providers (OMS-HC) [25] working in PHC environments in Brazil. Our aim is to validate and estimate the reliability of the OMS-HC scale using a sample of subjects from Brazil by examining the psychometric properties through the analysis of its internal consistency and factorial structure. Our study is among the first to explore the OMS-HC using a bifactor model, which allows for a deeper exploration of the psychometric properties of the dimensionality and reliability of the scale and its subscales.

## 2. Materials and Methods

### 2.1. Background

The current study is part of a cross-sectional project designed to investigate the presence of stigma toward people with mental illness and substance use problems in PHCs in Brazil (“Exploring Stigma, Discrimination and Recovery-Based Perspectives toward Mental Illness and Substance Use Problems among Primary Health care Providers in Ribeirao Preto, Brazil: A Randomized Controlled Trial”). The project is being implemented in Family Health Units (FHU), a public health unit designed to provide continuous care in basic specialties, with a multidisciplinary team qualified to carry out promotion, protection and recovery activities, characteristic of the primary care level, through the Family Health Strategy (FHS).

### 2.2. Participants

The participants, professionals who are part of family health teams, were recruited between June and November 2019, in Ribeirao Preto, Sao Paulo State, Brazil. All professionals from the six Family Health Units were invited to participate. Each FHS team included a nurse, a nurse assistant, a physician, and four to six full-time community health workers. The inclusion criterion was working time at the health unit of more than 1 month. As the FHTs were composed of different numbers of professionals in each professional category, the invitation to participate in the study was extended to all professionals, in order to have a representativeness of each category considering the six FHUs. The study was presented by the researchers to each professional individually and, to those who agreed to participate, a Free and Informed Consent Form was presented. The professionals answered the scale at their FHU, at a time when they were not responsible for the servicing (welcoming, consultation) of the users. It is noteworthy that among the professionals that make up the FHS are Community Health Workers, who are important PHC providers and are considered to be effective because they are part of the communities in which they work. Community health workers composed 50% of the professionals in the family health teams.

### 2.3. Measures

The OMS-HC was culturally adapted to the Brazilian context through a multi-step process. First, the research team translated the original instrument from English to Brazilian Portuguese, which was evaluated by a committee of experts who are fluent in both languages. The next steps involved back-translation and a pre-test. A pre-test of the preliminary instrument was performed from July to August of 2018 with a total of 40 health professionals from seven Primary Care Health Units in the city of Ribeirao Preto. Based on the pre-test results, the Brazilian version of the OMS-HC was determined to possess appropriate language that was easy to understand, good formatting that allowed ease of use, and appropriate consistency in relation to the original version [29].

The culturally adapted OMS-HC was used to collect data regarding validation, since the use of the scale depends on the assessment of the psychometric properties. Cultural adaptation does not validate a scale until the psychometric properties of the scale are confirmed [30,31]. The data collection for the validation of the culturally adapted OMS-HC scale was carried out from June to November 2019, and was conducted in person using a cross-sectional design by a research team who had been previously trained to administer the instrument.

The Mental Illness: Clinician’s Attitude Scale (MICA) measures clinicians’ attitudes towards people with mental illness, and was also used to further validate the culturally adapted version of the OMS-HC. The convergent validity of the OMS-HC scale was assessed by calculating the Spearman correlation between the total scores for the OMS-HC and MICA-4. A 95% confidence interval was estimated for the correlation using 1000 bootstrap samples. It is important to note that the MICA scale has not yet been validated for the Brazilian context. MICA-4 is a self-report questionnaire that contains 16 items and measures five dimensions related to stigma, views from the fields of social care, health and mental illness, knowledge about mental illness, disclosure, distinction between physical and mental health, and care of the patient with mental illness [32].

### 2.4. Statistical Analysis

Initial data cleaning and inspection were conducted, and the items that were negatively worded had their score reverse-coded to facilitate the interpretation of results. One respondent was removed due to missing values in all items. Most items had no missing values, while a few items had 1 or 2 missing values. These missing values were included in the data set and were either dealt with via a full maximum likelihood estimation or pairwise deletion, depending on the analysis.

Initially, the dimensionality of the scale was explored using Exploratory Factor Analysis (EFA) with weighted least square and mean and variance-adjusted chi-square test (WLSMV) estimation, which uses polychoric correlation and is appropriate for ordinal items [33]. Geomin oblique rotation [34] was used for the interpretation of the factors loadings. The number of factors was determined based on parallel analysis and factor interpretation considering 1 to 7 factor solutions. The parallel analysis calculated eigenvalues from 1000 synthetic correlation matrices from random data (i.e., no factor structure) to compare with the actual eigenvalues from the real data [35].

Confirmatory Factor Analyses (CFA) with different specifications (bifactor model with 4 specific and 1 general factor, CFA with 1, 4 and 5 specific factors, and a second-order factor model with 4 specific factors) were also used to evaluate the OMS-HC construct. Fit indices were reported, among them: CFI (Comparative Fit Index), TLI (Tucker–Lewis index), RMSEA (Root Means Square Error of Approximation), SRMR (Standardized Root Mean Square Residual), AIC (Akaike Information Criteria), BIC (Bayesian Information Criteria), SABIC (Sample Size Adjusted BIC) and the Chi-square test [36,37,38].

The dimensionality of the data was explored using a bifactor model and derived ancillary measures, as well as comparison with confirmatory factor analysis [39]. Because of the more complex nature of the bifactor model estimated with ordinal data, the WLSMV estimator did not work well, and this analysis used Robust Maximum Likelihood estimation (MLR), which assumes continuous data and uses robust estimates of standard errors equivalent to the Huber–White sandwich method [40]. A sensitivity analysis was performed, comparing the MLR and WLSMV estimators for a Confirmatory Factor Analysis with 16 items (See Figure A1), and they seemed to both result in similar sizes and rank orders of coefficients. However, the standard errors were consistently larger for the MLR, which is probably due to its robust nature. This should not be consequential for the conclusions given that we do not rely on *p*-values and statistical tests. Similar values for model fit indices also became apparent in our analysis.

The use of the bifactor model for the psychometric evaluation of scales has increased in the literature, particularly regarding the dimensionality of the scale and the relevance of subscales [41]. The bifactor model, if assumed to be correct, allows the calculation of a range of ancillary psychometric measures that are relevant for the understanding of the dimensionality and reliability of the OMS-HC scale. Measures such as Explained Common Variance (ECV), Individual Explained Common Variance (I-ECV), Percent of Uncontaminated Correlation (PUC) and Average Relative Parameter Bias (ARPB) are measures of dimensionality. Coefficient Omega (ω), Omega Hierarchical (ωH), Omega Hierarchical Subscale (ωHS) and Percent of Reliable Variance (PRV) are model-based reliability measures. We describe them shortly in Appendix B. Both types of measure are important to understanding the extent to which unit-weight total scores created from individual items represent the construct of interest (general OMS-HC factors), and the extent to which the subscales (specific OMS-HC factors) are different from each other and from the general factor. Although the bifactor is our primary model, we also present results from confirmatory factor analysis with 1, 4 and 5 factors, as well as a second-order factor analysis, since the choice of the best model can be controversial. All analyses have been conducted using software Mplus 8.2 [42].

### 2.5. Ethical Aspects

The study was approved by the Research Ethics Committee of the Ribeirão Preto College of Nursing (CAAE: 26431119.6.0000.5393). Participants who agreed to participate in the study read and signed the Free and Informed Consent Form.

## 3. Results

### 3.1. Sociodemographic Characteristics

There were 195 participants in total; 55% (107) were health care professionals and 45% (88) were community health workers, who are recognized as playing a central role in communication between the community and health services. The health professionals mostly comprised nurses (63%), but also included other health care providers, such as managers, pharmacists, and dentists. The community health workers included in the study were all nurses. The average participant age was 45.0 years (SD = 9.5), and 88% were females (Table 1).

### 3.2. Validity

Evidence was also found that the OMS-HC scale was robust to social desirability biases. The translation of the scale for use in the Brazilian context involved a range of procedures that sought to preserve these characteristics. Among them, health care experts and practitioners reviewed the translated item wording to ensure that the content of the items and its relation to stigma was valid for the Brazilian context. Bilingual academics as well as professionals experienced with the use of the scale in Canada were involved to avoid loss of content. In addition, convergent validity was assessed by the correlation between the 16-items OMS-HC scale and the MICA scale, with which OMS-HC is considered conceptually close. A Spearman correlation coefficient of 0.64 (95% Bootstrap CI: 0.55–0.72) was estimated, which seems reasonable.

### 3.3. Individual Item Analysis

The Cronbach’s Alpha was estimated to be 0.73, an acceptable level for the 20 items in the OMS-HC. However, four individual items (2, 5, 11 and 15) possessed correlation scores lower than 0.2 with the total scale; three of these were found to increase the alpha if removed (Table A1). Upon reviewing the specified items, all four were removed from the scale as they seemed to have been inconsistently interpreted by respondents. As a result, the alpha increased from 0.73 to 0.74 for the 16 remaining items. The psychometric properties of these 16 items were then assessed.

### 3.4. Dimensionality

The EFA conducted with 16 items resulted in four factors (See Table 2). This is the number of factors suggested by the parallel analysis and also by interpretation. Other criteria, such as eigenvalue larger than 1 or significance of the Chi-square difference test, suggested more than four factors, but such a large number of factors was inconsistent with the conception of the scale and the existing literature. A study in Hungary [43] found three factors, which is not entirely inconsistent with our findings (Table 2), but four factors is meaningfully more than was reported in the original OMS-HC development paper, where two factors were found using 12 items. Our three-factor solution showed several cross-loadings, and the factors did not yield a clear interpretation. When adding a fourth factor, the Disclosure factor emerged, and in our evaluation, this was found to be consistent and important for the Brazilian context. Based on the interpretation of the items, the first factor was named “Disclosure”, as it is associated with the idea of comfort in disclosing one’s mental health status. The second factor, “Social Distance”, is related to aspects of a relationship with a person affected by mental health issues. The third factor was named “Attitudes” as it reflects attitudes and perceptions towards mental health. The fourth factor is labeled “Negative Views” and is related to negative attitudes from others towards those with mental health conditions. In comparison with the available literature, this last factor seems to stand out as different and may reflect specificities of the Brazilian culture. As we see in Table 2, some items load in more than one factor. For example, item 17 states “I would not want a person with a mental illness, even if it were appropriately managed, to work with children” primarily loads on “Negative Views”, but also involves “Social Distance”, and to a lesser extent, negatively loads on the “Attitudes” factor. This is not entirely inconsistent, and we decided to keep the item in the analysis.

### 3.5. The Bifactor Model

The path diagram for the bifactor model is presented in Figure A2. It models simultaneously the specific OMS-HC factors (OMS subscales) and the general OMS-HC factor, and in doing so, allows us to gain insights regarding the dimensionality and reliability of the OMS-HC total score and its subscales. The specific factors used were derived from the EFA, with the understanding of the limitations of using the sample for this analysis, that is, the fit is expected to be better than in external samples. In Table 3, fit indices are presented for different models, and the bifactor model along with the second-order factor model are shown to have the best fit to the data, while the bifactor model also has an adequate fit (RMSEA = 0.04 (95% CI: 0.016–0.059), CFI = 0.90, TLI = 0.87, and lower AIC and SABIC). This is important because the ancillary measures derived from the bifactor model assume that it represents the factor structure in the data sufficiently.

It is also important to notice in Table 3 that the bifactor model fits better than the one-factor CFA, which is the implied model used when total factor scores are used in practice. It also fits better than the CFA with four subscales, indicating that the general OMS-HC factor does seem to contribute beyond the specific factor contributions. The second-order factors also show a fit equivalent to the bifactor model. This model implies a general factor that is defined by the shared variance between the specific factors, therefore also implying that the specific factors are correlated. The bifactor model, on the other hand, assumes they are not correlated given the general factor, which encourages the use of bifactor models to compare the specific and general factors. The reasonable fit of the second-order factor is probably explained by the existence of a correlation between the specific factors found in the EFA (from 0.17 to 0.25, all of which were significant at *p* = 0.001 or lower).

Using the bifactor model, the ECV (Explained Common Variance) for the general OMS-HC factor was 0.42 (SE = 0.06). For the specific factors, the ECV was 0.15, 0.10, 0.18 and 0.15, respectively for Disclosure, Social Distance, Attitudes and Negative Views. The ECV for the general OMS factor, which was not so high, indicates that the variance in the total OMS-HC score does not come from a single dimension and, in fact, only 42% of it does. The remaining 58% of the variance explained is captured by aspects of the subscales.

The PUC (Percent of Uncontaminated Correlation) was found to be 0.78, which indicates that the majority of the information in the correlation matrix is about the general factor, and this may indicate that despite the moderate ECV, the use of the total OMS-HC score (that is, assuming the one-factor model) is probably close to what we would get with a bi-factor model, assumed to be the better specified model [39]. This result gives some foundation for the use of the OMS-HC total score as a sum of scores, as is done in practice.

**Table 3 healthcare-11-01049-t003:** Fit indices for the confirmatory factor models. Both the bifactor and the second-order factor models have 5 factors, which are the 4 specific factors and the general factor.

	4 Factors	4 Factors	5 Factors	1 Factor	5 Factors
	CFA	CFA	Bifactor	CFA	Second-Order
	20 Items	16 Items	16 Items	16 Items	16 Items
Chi-square	222.4	133.6	117.6	199.9	132.4
df	164	98	89	104	100
*p*-value	0.0016	0.0098	0.0229	<0.0001	0.0166
RMSEA	0.043	0.043	0.041	0.069	0.041
RMSEA CI	0.027–0.056	0.022–0.061	0.016–0.059	0.054–0.083	0.018–0.059
CFI	0.837	0.878	0.902	0.671	0.889
TLI	0.811	0.85	0.868	0.62	0.866
SRMR	0.067	0.06	0.057	0.076	0.06
AIC	10,514.842	8356.547	8353.067	8435.081	8352.578
BIC	10,730.521	8533.012	8558.942	8591.938	8522.507
SABIC	10,521.446	8361.951	8359.371	8439.884	8357.781

CFI = Comparative Fit Index. Values higher than 0.95 are considered adequate [12]. TLI = Tucker–Lewis index. Values higher than 0.95 are considered adequate [12]. RMSEA = Root Means Square Error of Approximation. A cut-off of 0.07 has been considered the upper limit for adequate fit [12]. SRMR = Standardized Root Mean Square Residual; 0.08 and lower are considered adequate [12]. AIC = Akaike Information Criteria. The lower the better [13]. BIC = Bayesian Information Criteria. The lower the better [14]. SABIC = Sample Size Adjusted BIC. The lower the better [15]. Chi-square = Chi-square Statistics for test of model fit. Lower *p*-values indicates stronger evidence of model mis-specification.

The I-ECV, which is the ECV at the item level, is shown in Table 4. It measures the proportion of the common variance in the item that reflects the general OMS-HC factor. We see that most items reflect the general OMS-HC factor more than the specific factors, but some items explain the specific factors more effectively (items 1, 9, 14, 17 and 19). The fact that the I-ECVs are not very high indicates multidimensionality in the OMS-HC scale. In Table 4, the percent bias for each item is a measure of disparity between the assumed better model (bifactor) and the one-factor model, which is used when a total score is calculated in practice. Averaging across all items results in an ARPB (Average Relative Percent Bias) of 34%, which is generally considered high. This indicates that using a unidimensional approach for the overall OMS-HC total scores is likely to produce biased scores relative to the total score from the bifactor model.

### 3.6. Reliability

The Cronbach’s Alpha for the OMS scale with 16 items was 0.74, which is considered to indicate reasonable reliability. The model-based reliability based on the bifactor model was ω = 0.78 (SE = 0.03). The reliability values for the specific factors were: Disclosure—ω = 0.91 (SE = 0.10); Social Distance—ω = 0.55 (SE = 0.06); Attitudes—ω = 0.68 (SE = 0.04); Negative Views—ω = 0.60 (SE = 0.05). While alpha uses a single-factor structure with equal loadings, omega uses the structure of the bifactor model, hence it is natural that the variance explained will be higher if we use omega.

While omega estimates the variance explained by all sources, the omega hierarchical looks at the variance explained by the factor of interest only. The omega hierarchical for the general OMS factor was ωH = 0.58 (SE = 0.07), indicating that 58% of the total variance in the total OMS score is due to a general OMS-HC factor (with around 20% due to specific factors and 22% being unique variance (variance specific to individual items or random)). This is an important measure because it shows that the unit-weight total OMS-HC score from the 16 items represents a general OMS-HC construct only to a moderate extent, with reasonable variation (42%) that is due to sources other than the general OMS factor.

The omega hierarchical values for each subscale were 0.11, 0.04, 0.25 and 0.2 for Disclosure, Social Distance, Attitude and Negative Views, respectively. Although the reliability of the subscales is not low, these values indicate that most of that reliability is due to the general OMS-HC factor and not to the specific factors. These results seem to indicate that the use of the total OMS-HC score should be done with some caution, and the use of subscales scores is probably not advised as they mostly reflect the overall factor, and represent few different unique constructs.

## 4. Discussion

In the original OMS-HC, the factor analysis identified two subscales that mediated attitudes towards people with mental illness, starting from a definition of stigma that included attitudes towards disclosing a mental illness and/or seeking help. The idea was adopted that stigmatizing attitudes can be measured in the form of disclosure if someone has a mental illness and/or is looking for help, as this reaction can also be an indicator of stigma related to mental illness [44,45]. On the other hand, the disclosure of a mental illness may not be associated with something shameful and, therefore, attitudes may be less stigmatizing towards other people. Thus, the factor analysis resulted in a subscale of seven items used to measure attitudes towards people with mental illness, and another of five items used to measure attitudes of disclosure/disclosure of a mental illness [29].

In Chile, the OMS-HC was confirmed with a three-factor structure and the mean score (α = 0.69) was 34.55 (theoretical range 15 to 75). In the Chilean validation, the Cronbach’s α for the three subscales was low, indicating that it is more appropriate, in the Chilean context, to use the full 15-item OMS-HC scale rather than individual subscales [28]. The Italian version of OMS-HC maintained the original structure, with satisfactory internal consistency. The scale consists of two subscales (attitudes of health professionals towards people with mental illness and attitudes of health professionals towards disclosing mental illness) for a total of 12 items [27].

While in the Brazilian version we also identified the two factors in the original version, more items were retained (16 instead of 12), and two additional factors were extracted. The results of this study corroborate more closely the findings from the Hungarian study, which analyzed the factor structure and psychometric properties of the Hungarian version of the OMS-HC. The Hungarian study also supported the theoretical approach of the scale that consists of Attitude, Disclosure and Help-Seeking, and Social Distance subscales [23]. The Brazilian version uncovered a new factor, “Negative Views”, which may be particular to the cultural context of Brazil.

Although we identified the results of other OMS-HC versions from other countries, it is important to emphasize that they concern different populations; therefore, the comparison results need to be interpreted with caution.

Our results show evidence for the multidimensionality of the OMS-HC scale, as seen by the factor analysis resulting in more than one factor, the unidimensional CFA performing worse than the four-factor CFA and bifactor models, and the low Explained Common Variance (ECV) and ARPB. These proofs of multidimensionality give justification for the bifactor analysis, and its assessment of the reliability and uniqueness of the general and specific factors.

In terms of reliability, the omega coefficient for the general OMS-HS construct from the bifactor model was high (0.78). It was also high for the subscales (0.6 or higher). The omega hierarchical, which categorizes the source of variance due to each factor, was moderate for the general scale (0.58) and low for the subscales (0.25 or lower), indicating that the total OMS-HC score may represent a general OMS construct reasonably; however, the subscales’ scores provide limited representation of different constructs, and reflect more of the overall OMS construct. This indicates that the use of subscales to measure specific constructs is probably not advisable.

A limitation of this study is the specific sample of participants, who work in primary health care in a certain municipality in Brazil. Given the sample, it is possible that the results may not be generalizable to other health professionals who work in health care institutions, which are vastly different from primary health care located in other parts of the country. In addition, the bifactor model used for psychometric analysis seems to have a reasonable fit, but may still not be the correct or ideal model for the OMS-HC scale. The Explained Common Variance for the general factor is low, indicating that caution must be used when interpreting the total OMS-HC score. The reliability of the general factor was also not very high (58%), again indicating that the total OMS-HC score has a substantial source of variance that does not come from the general factor. We also recognize the sample size limitation, which did not allow us to apply the bifactor model to an independent sample from that used for the exploratory factor analysis, which may have resulted in overestimated fit indices.

## 5. Conclusions

The data presented on the factorial structure and psychometric properties of the Brazilian version of OMS-HC contribute to the use of the scale in studies that investigate the level of stigma among health professionals in relation to people with mental illness, as well as in the development of anti-stigma interventions in this context. The monitoring of stigmatizing attitudes of health professionals may be permitted through the OMS-HC, strengthening the implementation of mental health policies. Additionally, the OMS-HC can contribute to the evaluation of the effectiveness of the actions of health professionals in relation to people with mental illness. Thus, the validation of the OMS-HC scale may not only help address the problem of estimating the prevalence of stigma, but also be a catalyzer for actions aiming at countering it.

## Figures and Tables

**Table 1 healthcare-11-01049-t001:** Characteristics of the sample.

		N	%
Total		195	100.0%
Gender	Female	172	88.2%
	Male	23	11.8%
Occupation/Function in the Unit (grouped)	Nurse	15	7.7%
	Nursing Assistant	39	20.0%
	Nurse Technician	11	5.6%
	Community Wealth Workers	88	45.1%
	Other	42	21.5%
Postgraduate	Yes	10	5.1%
	No	185	94.9%
Know someone close to you with a mental disorder	Yes	177	91.2%
	No	17	8.8%
Have you ever cared for/treated someone with a mental disorder	Yes	130	67.4%
	No	63	32.6%
Age (Mean/SD)		44.9	9.5
Questionnarie time in minutes (Mean/SD)		10.4	10.9
Time working in years (Mean/SD)		18.1	9.4
Time working in the health unit in months (Mean/SD)		83.1	59.5

Note: The sample size is 195, but some questions relate to fewer participants because of missing values.

**Table 2 healthcare-11-01049-t002:** Standardized factor loadings for Exploratory Factor Analysis with 4 factors estimated using WLSMV method and GEOMIN rotation. “*” indicates statistical significance. We focus the interpretation of the factors on the loadings that are higher than 0.3 or the loading that is the highest for the item.

	Disclosure	Social Distance	Attitude	Negative View
4. If I were under treatment for a mental illness I would not disclose this to any of my colleagues.	**0.678 ***	0.206 *	−0.118 *	−0.012
10. If I had a mental illness, I would tell my friends.	**0.847 ***	−0.08	0.110 *	0.085
3. If a colleague with whom I work told me they had a managed mental illness, I would be as willing to work with him/her.	0.294 *	**0.510 ***	0.024	0.139 *
8. Employers should hire a person with a managed mental illness if he/she is the best person for the job.	−0.02	**0.645 ***	0.108	0.005
9. I would still go to a physician if I knew that the physician had been treated for a mental illness.	0.157 *	**0.495 ***	0.004	0.169 *
19. I would not mind if a person with a mental illness lived next door to me.	0.013	**0.462 ***	0.213 *	0.057
7. I would be reluctant to seek help if I had a mental illness.	0.135	0.220 *	**0.407 ***	−0.033
1. I am more comfortable helping a person who has a physical illness than I am helping a person who has a mental illness.	0.035	0.062	**0.276 ***	0.089
6. I would see myself as weak if I had a mental illness and could not fix it myself.	0.085	0.092	**0.498 ***	0.04
12. Despite my professional beliefs, I have negative reactions towards people who have mental illness.	0.159 *	0.108	**0.541 ***	−0.001
20. I struggle to feel compassion for a person with a mental ilness.	−0.023	−0.044	**0.588 ***	0.074
13. There is little I can do to help people with mental illness.	0.277 *	−0.163 *	0.088	**0.605 ***
14. More than half of people with mental illness don’t try hard enough to get better.	−0.148	0.041	0.248 *	**0.375 ***
16. The best treatment for mental illness is medication.	−0.03	0.004	0.016	**0.523 ***
17. I would not want a person with a mental illness, even if it were appropriately managed, to work with children.	0.014	0.337 *	−0.166 *	**0.585 ***
18. Health care providers do not need to be advocates for people with mental illness.	0.103	0.004	0.197 *	**0.260 ***

Note: Bold values indicate the items loadings considered in each factor; asterisks indicates significant loadings at *p* < 0.05.

**Table 4 healthcare-11-01049-t004:** Standardized item loadings for each model. Bias = difference in percent between loadings in the general OMS factor from the bifactor model and the CFA with a single factor. IECV = ECV at the item level; percent of the item common variance that is explained by the general factor.

Factors	Item	Bifactor		CFA Loadings	1-Dim. CFA Loadings	Second-Order Factor Loadings	Bias	IECV
		Gen. Factor Loadings	Spec. Factor Loadings					
Disclosure	4	0.456	0.584	0.724	0.506	0.723	11%	84%
	10	0.399	0.616	0.748	0.495	0.749	24%	58%
Social Distance	3	0.561	0.665	0.625	0.486	0.624	13%	62%
	8	0.454	0.152	0.518	0.322	0.519	29%	76%
	9	0.669	−0.191	0.527	0.495	0.527	26%	33%
	19	0.376	0.224	0.474	0.36	0.475	4%	10%
Attitudes	1	0.176	0.400	0.410	0.304	0.409	73%	8%
	7	0.299	0.210	0.390	0.399	0.391	33%	70%
	6	0.263	0.467	0.552	0.397	0.552	51%	75%
	12	0.304	0.521	0.563	0.354	0.563	16%	67%
	20	0.097	0.377	0.369	0.156	0.369	61%	61%
Negative Views	13	0.306	0.438	0.523	0.423	0.525	38%	88%
	14	0.197	0.249	0.351	0.256	0.350	30%	30%
	16	0.200	0.536	0.456	0.29	0.456	45%	79%
	17	0.494	0.319	0.599	0.474	0.598	4%	26%
	18	0.161	0.309	0.335	0.297	0.335	84%	94%

## Data Availability

The study database is on file in Google Drive and can be requested from the self-correspondent via email.

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
