# Peer review of "Validation of the Opening Minds Scale for Health Care Providers (OMS-HC): Factor Structure and Psychometric Properties of the Brazilian Version"

_healthcare, 2023, doi:10.3390/healthcare11071049_

Round 1

Author Response

Dear reviewer,

Thank you for the suggestions regarding our article, aiming at improving it and making it suitable for publication. We reorganized the article in order to address all comments and suggestions. However, we are willing to continue working on it, if you think it still needs improvement.

Reviewer 2 Report

It has been very interesting for me to review this article. I think that the WHO HC scale is a widely used scale and it is necessary to have an instrument that helps us to measure the stigma present in health professionals.

I would like to make some comments that I think would improve the quality and legibility of the article

Introduction

The introduction does not discuss the original instrument, its psychometric properties and the importance of translating this particular instrument.

Materials and method, 

In my opinion, lines 68-81 on materials and methods could be deleted as they do not provide information that readers do not already know.

Reliability 

I would eliminate the explanation of the use of the omega coefficient, providing only the data obtained. 

In the Statistical Analysis section, I would justify the omega coefficient with bibliography.

Discussion

I believe that the results obtained should be compared with other adaptations that have been made of the instrument.

Author Response

(The authors gave the same response as above.)

Reviewer 3 Report

Thank you very much for the opportunity to review this manuscript. The manuscript covers a very important area - healthcare provider attitudes toward mental illness - and provides research on a Brazilian version of the OMS-HC. I thought the manuscript was well-written overall. Here is my feedback, which I hope the authors find helpful:

It would be useful to write in the Introduction about the psychometric results of the OMS-HC questionnaire. For example, it could be shown in which countries (e.g., Canada, Chile, Hungary, Italy) the factor structures were obtained. For example, the following statement is not true, since the authors used a bifactor model when examining the Hungarian factor structure of OMS-HC: „Our study is among the first to explore at the OMS-HC scale using a bifactor model, which allows for a deeper exploration of the psychometric properties of dimensionality and reliability of the scale and its subscales.

It would be worth presenting the process of developing the original scale and embedding this in the interpretation of your own results, e.g., the initial version of the OMS-HC (20 items) and the 15-item version of the scale.

The Materials and Methods section contains the template text of the journal: „The Materials and Methods should be described with sufficient details to allow others to replicate and build on the published results….”

The individual item analysis is not the most appropriate method to shorten the original 20 items, i.e., to omit items 2, 5, 11, and 15. It would therefore be useful to present the original and abbreviated versions of the OMS-HC in light of the psychometric analyses in the introduction.

The following wording in the Statistical analysis section is inaccurate: „A bifactor model and Confirmatory Factor Analyses (CFA) were also used to evaluate the OMS-HC construct.” The bi-factor model is a confirmatory factor analytic model originally proposed for measurement data by Holzinger and Swineford (Holzinger & Swineford, 1937).  

I think that the results of the EFA analysis are definitely worth reconsidering. Although the result of the parallel analysis indicated a 4-factor solution, it might be useful to check the 3-factor solution in light of international results and your own CFA results. Items 4 and 10 (4. If I were under treatment for a mental illness, I would not disclose this to any of my colleagues; 10. If I had a mental illness, I would tell my friends.) may not even be considered independent and meaningful factor, but rather as item doublet, characterized by a very close correlation between two similarly worded items. For more details, see for example Ferrando et al., 2022. The interpretation of these two items as independent factors causes issues in the final scale structure construction. Accordingly, I also do not feel justified in the following statement from the conclusion: "The Brazilian version uncovered a new factor, Negative Views which may be particular to the cultural context of Brazil."

When analyzing different confirmatory factor analysis (CFA) models, the fit indices are low, and the bifactor design is not really comparable to other designs because it usually gives a better fit. For more details, see for example Reise et al., 2018.  

In conclusion, I feel that the manuscript is important and valuable, however, I suggest that both the EFA and CFA models should be reconsidered as the fit index of the adopted bifactor model is very low.    

References

Ferrando, P. J., Hernandez-Dorado, A., & Lorenzo-Seva, U. (2022). Detecting Correlated Residuals in Exploratory Factor Analysis: New Proposals and a Comparison of Procedures. Structural Equation Modeling: A Multidisciplinary Journal, 1-9.

Holzinger, K. J., & Swineford, F. (1937). The bi-factor method. Psychometrika, 2, 41–54. https://doi.org/10.1007/BF02287965

Reise, S. P., Bonifay, W., & Haviland, M. G. (2018). Bifactor modelling and the evaluation of scale scores. The Wiley handbook of psychometric testing: A multidisciplinary reference on survey, scale and test development, 675-707.

Author Response

(The authors gave the same response as above.)

Round 2

Reviewer 1 Report

1. The acronym CHA in Table 1 should be spelled out.

2. It is important to recruit the representative subjects in the specific field, when the clinical scale is examined for the validation. The authors mentioned that they made an effort to give all professionals in the region an opportunity to participate, including by making their participation easier and by being flexible on contact time and sending reminders. It would be worth presenting the process of their recruiting the subjects in the method section.

Author Response

Dear reviewer,

Thank you for the suggestions regarding our article, aiming at improving it and making it suitable for publication.

Reviewer 2 Report

Thank you for allowing this manuscript to be reviewed again.

In the previous comments, I suggested to the authors that they should include in the introduction information about the importance of the questionnaire they are going to validate and its use in other countries. Perhaps the description they have made this time is too long and they provide data that are not relevant in this section and which makes the introduction section too long, so I would delete from line 70 to the point followed by line 83.

As for the rest of the article, I believe that the modifications that have been made have improved its quality. Congratulations to the authors

Author Response

(The authors gave the same response as above.)

Reviewer 3 Report

Thank you for your careful revision, I fully agree with the answers.

Author Response

(The authors gave the same response as above.)
